# Triplet–triplet upconversion enhanced by spin–orbit coupling in organic light-emitting diodes

Ryota Ieuji[1,2], Kenichi Goushi[1,2,3]* & Chihaya Adachi [1,2,3]*

Triplet–triplet upconversion, in which two triplet excitons are converted to one singlet exciton, is a well-known approach to exceed the limit of electroluminescence quantum efficiency in conventional fluorescence-based organic light-emitting diodes. Considering the spin multiplicity of triplet pairs, upconversion efficiency is usually limited to 20%. Although this limit can be exceeded when the energy of a triplet pair is lower than that of a second triplet excited state, such as for rubrene, it is generally difficult to engineer the energy levels of higher triplet excited states. Here, we investigate the upconversion efficiency of a series of new anthracene derivatives with different substituents. Some of these derivatives show upconversion efficiencies close to 50% even though the calculated energy levels of the second triplet excited states are lower than twice the lowest triplet energy. A possible upconversion mechanism is proposed based on the molecular structures and quantum chemical calculations.

[1] Department of Applied Chemistry and Center for Organic Photonics and Electronics Research (OPERA), Kyushu University, 744 Motooka, Nishi, Fukuoka 819-0395, Japan. [2] Japan Science and Technology Agency (JST), Exploratory Research for Advanced Technology (ERATO), Adachi Molecular Exciton Engineering Project, Kyushu University, 744 Motooka, Nishi-ku, Fukuoka 819-0395, Japan. [3] International Institute for Carbon Neutral Energy Research (WPI-I2CNER), Kyushu University, 744 Motooka, Nishi-ku, Fukuoka 819-0395, Japan. *email: goushi@opera.kyushu-u.ac.jp; adachi@opera.kyushu-u.ac.jp

Recently, organic light-emitting diodes (OLEDs)[1] have been widely used in practical applications such as solid-state lighting with low-power consumption[2,3] and high-resolution flexible displays. In organic semiconductors, injected electrons and holes from opposite electrodes recombine to form singlet excitons ($S_1$) with a probability of 25% and triplet excitons ($T_1$) with a probability of 75%[4]. Therefore, efficient OLEDs require the contribution from $T_1$. In OLEDs containing transition metal complexes as emitters, $T_1$ contribute directly to electroluminescence (EL) because heavy atom effects can realize efficient room-temperature phosphorescence through strong spin–orbit coupling[5], leading to nearly 100% internal EL quantum efficiency ($\eta_{IQE}$)[6]. Alternatively, we recently found that OLEDs with emitters that utilize thermally activated delayed fluorescence (TADF)[7], in which $T_1$ can be converted into $S_1$ through reverse intersystem crossing (RISC) at room temperature, achieved $\eta_{IQE}$ close to 100%[8,9]. Although OLEDs that use room-temperature phosphorescence and TADF can harvest triplet excitons efficiently, it is still challenging to realize deep blue OLEDs using $T_1$[10–12]. This is because a simplified molecular architecture has to be used to obtain high $T_1$ energy, which restricts the molecular design for both phosphorescence and TADF materials. To escape these problems, triplet–triplet upconversion (TTU), in which low $T_1$ energy can be upconverted to high $S_1$ energy close to twice the $T_1$ energy, can expand the scope for molecular design[13–20].

When a migrating $T_1$ collides with another $T_1$, they form an intermediate state. According to spin statistics with the strong-coupling limit, the spin multiplicity leads to the formation of singlet intermediate states ($^1(TT)$) with a probability of 1/9, triplet intermediate states ($^3(TT)$) with a probability of 1/3, and quintet intermediate states ($^5(TT)$) with a probability of 5/9 (refs. [14,16,21]). The $^1(TT)$ can be converted into $S_1$ and a molecular ground state ($S_0$), and the $^3(TT)$ can be converted into $T_1$ and $S_0$. Conversely, the $^5(TT)$ returns to two $T_1$, because the quintet (intramolecular) excited state ($Q_1$) formed by two-electron excitations, which is another possible path, is appreciably higher in energy than $^5(TT)$[22]. Therefore, assuming that both $^1(TT)$ and $^3(TT)$ are completely converted into $S_1$ and $T_1$, respectively, without returning to two $T_1$, the total TTU efficiency ($\eta_{TTU}$) from one $T_1$ to one $S_1$ by repeated TTU is given by Eq. (1)[14,16]:

$$\eta_{TTU} = \sum_{k=0}^{\infty} \frac{1}{18} \times \left(\frac{13}{18}\right)^k = 20\%. \tag{1}$$

Based on the maximum $\eta_{TTU}$ of 20%, the theoretical limit of the external EL quantum efficiency ($\eta_{EQE}$) utilizing TTU is given by Eq. (2):

$$\eta_{EQE} = \gamma_{carrier} \times \eta_r \times \Phi_{PL} \times \eta_{OC} \approx 1.0 \times (0.25 + 0.75 \times 0.2) \times 1.0 \times 0.2 = 8\%, \tag{2}$$

where $\gamma_{carrier}$ is the carrier balance factor, $\eta_r$ is the radiative-exciton production efficiency, $\Phi_{PL}$ is the photoluminescence (PL) quantum yield, and $\eta_{OC}$ is the outcoupling efficiency. Assuming an $\eta_{OC}$ of 20%, the maximum $\eta_{EQE}$ is limited to 8%.

Alternatively, it has been suggested that the limit of $\eta_{TTU}$ can be exceeded when the energy level of $^3(TT)$ is lower than that of a second triplet state ($T_2$)[16,23]. Under this condition, $^3(TT)$ can be expected to return to two $T_1$, because the transition from $^3(TT)$ to one $T_1$ requires an instantaneous release of the $T_1$ energy as vibronic energy. As a result, the maximum $\eta_{TTU}$ can be raised to 50% because of the recycled $T_1$, leading to the higher theoretical limits of $\eta_\gamma = 62.5\%$ and $\eta_{EQE} = 12.5\%$. However, the superior TTU is limited to a few materials such as rubrene (5,6,11,12-tetraphenylnaphthacene)[13,15,19] and an anthracene derivative[24] because of the tight restriction of the $T_2$ energies. As another

approach exceeding the $\eta_{TTU}$ limit, the transition from $^3(TT)$ (or a higher triplet state, $T_n$) to $S_1$ has been suggested[23]. Although this transition is an attractive path to improve $\eta_{TTU}$ because there is no tight restriction of the $T_2$ energy, the factors controlling this transition are still unclear.

In this study, we evaluate $\eta_{TTU}$ of anthracene derivatives with different electron-donating and -withdrawing substituents. We show that a few anthracene derivatives show conversion efficiencies close to 50%, indicating an upconversion mechanism involving the transition from $^3(TT)$ to $S_1$. Based on the molecular structures and results of quantum chemical calculations, we propose an upconversion mechanism involving spin conversion for these materials.

## Results

**Molecular design and synthesis.** Figure 1a shows the general TTU mechanism. Figure 1b shows the chemical structures of the investigated TTU materials. DMAC-σ-ANT (9,10-dihydro-9,9-dimethyl-10-(4-(10-phenylanthracen-9-yl)phenyl)acridine), DMAC-σ-ANTCN (4-(10-(4-(9,9-dimethylacridin-10(9H)-yl)phenyl)anthracen-9-yl)benzonitrile), PXZ-σ-ANT (10-(4-(10-phenylanthracen-9-yl)phenyl)-10H-phenoxazine), PXZ-σ-AN-TCN (4-(10-(4-(10H-phenoxazin-10-yl)phenyl)anthracen-9-yl)benzonitrile), and ANT-TRZ (2,4-diphenyl-6-(4-(10-phenylanthracene-9-yl)phenyl)-1,3,5-triazine) were synthesized. The intermediate 4-[(10-(4-chlorophenyl)-9-anthracenyl]benzonitrile was synthesized according to the reported methods[17] and all materials were purified by vacuum sublimation. The synthetic route and detailed synthesis of these materials are provided in Supplementary Note 1. The structures of these materials were fully characterized by $^1H$ and $^{13}C$ nuclear magnetic resonance (NMR) spectroscopies and elemental analyses.

We introduced different electron-donating and -withdrawing substituents into 9,10-diphenylanthracene (DPA) to give materials with the common skeleton of an anthracene unit, which is well known to provide efficient TTU[25]. DMAC-σ-ANT and DMAC-σ-ANTCN contain dimethylacridine and PXZ-σ-ANT and PXZ-σ-ANTCN contain phenoxazine as electron-donating substituents. The electron-donating ability of phenoxazine is stronger than that of dimethylacridine. We designed these four anthracene derivatives such that the core anthracene units provided electron-withdrawing substituents. Moreover, to enhance the electron-withdrawing ability of some of the materials, we introduced a cyano group onto the anthracene units in DMAC-σ-ANTCN and PXZ-σ-ANTCN. ANT-TRZ contains 2,4,6-triphenyl-1,3,5-triazine as an electron-withdrawing substituent and anthracene as an electron-donating substituent. Supplementary Figure 1 and Supplementary Table 1 show the molecular orbitals and oscillator strengths, respectively, of the anthracene derivatives calculated by time-dependent density function theory (TD-DFT) calculations (B3LYP/6-31G(d)).

**Photophysical properties.** Figure 2 shows ultraviolet–visible (UV–Vis) absorption and PL spectra of thin films of the anthracene derivatives. The absorption spectra contain vibronic structures originating from the anthracene units and no charge transfer (CT) absorptions. These results agree well with the small oscillator strengths of $S_1$ calculated by the TD-DFT calculations. Featureless PL spectra were obtained for DMAC-σ-ANTCN, PXZ-σ-ANT, PXZ-σ-ANTCN, and ANT-TRZ, indicating each emission originates from CT. The PL spectrum of DMAC-σ-ANT consisted of a broad band superimposed with vibronic structures, suggesting a mixture of the singlet localized excited state ($^1$LE) and singlet CT state ($^1$CT). The PL peak wavelengths of DPA, DMAC-σ-ANT, DMAC-σ-ANTCN, PXZ-σ-ANT, PXZ-σ-

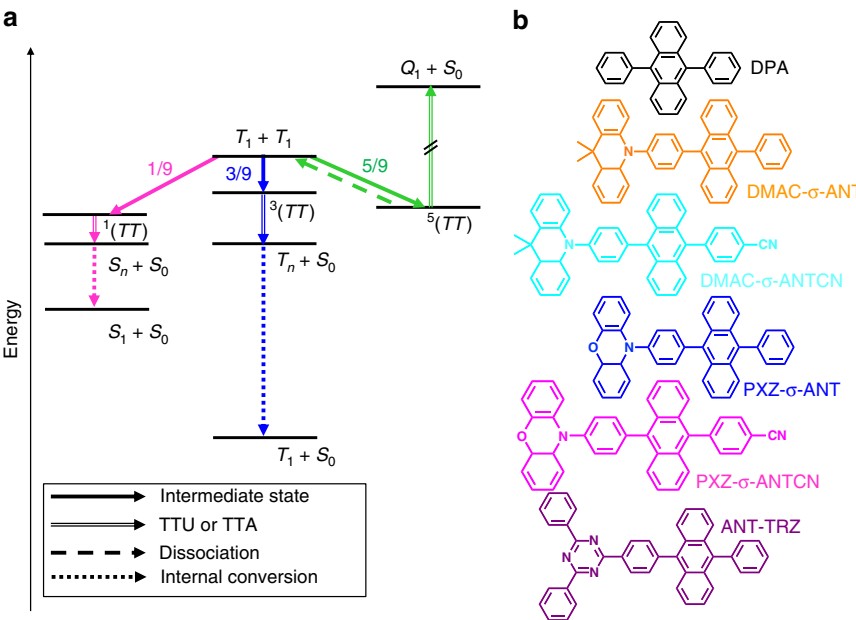

**Fig. 1** Conventional triplet-triplet upconversion mechanism. **a** Energy-level diagram illustrating the mechanisms of triplet–triplet upconversion (TTU) and triplet–triplet annihilation (TTA). The $^1(TT)$, $^3(TT)$, and $^5(TT)$ are singlet, triplet, and quintet intermediate states. The $S_1$, $T_1$, and $S_0$ are singlet excitons, triplet excitons, and ground states. The $S_n$ and $T_n$ are higher singlet and triplet states. **b** Chemical structures of the anthracene derivatives.

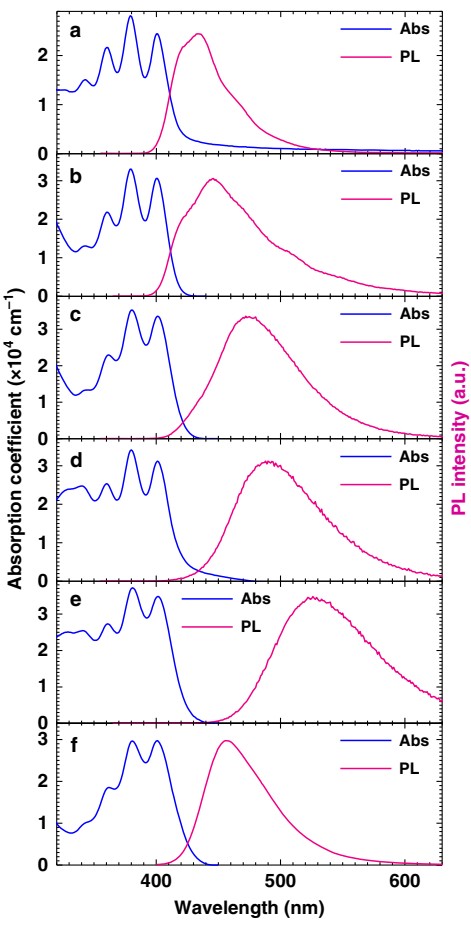

**Fig. 2** Absorption and photoluminescence of thin films. UV-Vis absorption and PL spectra of thin films of **a** DPA, **b** DMAC-σ-ANT, **c** DMAC-σ-ANTCN, **d** PXZ-σ-ANT, **e** PXZ-σ-ANTCN and **f** ANT-TRZ.

ANTCN, and ANT-TRZ are 434 nm (2.85 eV), 446 nm (2.78 eV), 472 nm (2.63 eV), 488 nm (2.54 eV), 526 nm (2.36 eV), and 456 nm (2.72 eV), respectively. To analyse the excited states, we investigated the absorption and PL spectra of the anthracene derivatives in different solvents as shown in Fig. 3a. According to the Lippert–Mataga equation[26], the dipole moment of the $S_1$ state can be extracting from a slope of the Stokes shift as a function of the orientation polarizability, $f$, as shown in Supplementary Table 2. The fitting results by Eq. (7) in Methods are shown in Fig. 3b, respectively. The estimated dipole moments of the $S_1$ for DPA, PXZ-σ-ANT, PXZ-σ-ANTCN, and ANT-TRZ are 2.1, 20.7, 21.3, and 16.0 D, respectively. The estimated dipole moments higher than ~10 D indicate that the excited states should be assigned to CT states. Therefore, these results agree with the assignments in the PL spectra. On the other hand, DMAC-σ-ANT and DMAC-σ-ANTCN show the two independent slops, indicating the two-different excited state altered by surrounding environments. In apolar solvents, the estimated dipole moments of the $S_1$ for DMAX-σ-ANT and DMAC-σ-ANTCN are 2.7 and 2.8 D indicating the excited states assigned to LE states, respectively. In polar solvent, those are 20.7 and 23.4 D, indicating the excited states assigned to CT states, respectively. The observed PL peaks of the DMAX-σ-ANT and DMAC-σ-ANTCN films are located between the results in apolar and polar solvents. Therefore, the $S_1$ states of the DMAX-σ-ANT and DMAC-σ-ANTCN films would be assigned to the mixture of the $^1$LE and $^1$CT, respectively. Although DMAX-σ-ANT agrees with the assignment in the PL spectra, the Lippert–Mataga plot revealed the mixture of the excited state for the DMAC-σ-ANTCN film. However, the PL spectrum of the DMAC-σ-ANTCN films is close to the PL spectra in polar solvents rather than that in apolar solvents. Therefore, the $S_1$ state of DMAC-σ-ANTCN in the film would be assign to a $^1$CT weakly mixing $^1$LE.

Figure 4 shows phosphorescence spectra of the anthracene derivatives (2 wt%) dispersed in a polymer matrix[27]. The derivatives show nearly identical phosphorescence spectra, indicating the lowest triplet excited state is the triplet localized excited state ($^3$LE) of the anthracene unit in all cases. Figure 5

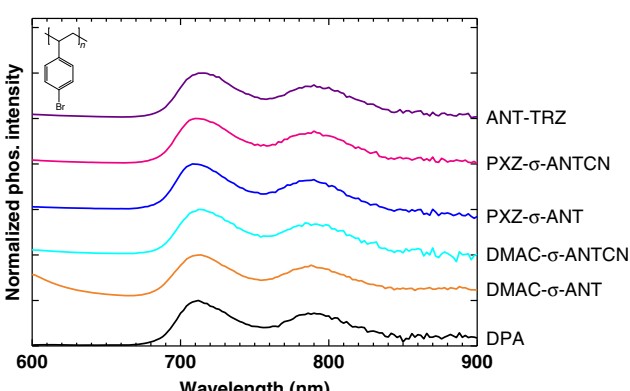

**Fig. 3** Solvent effects. **a** Absorption and fluorescence spectra of the anthracene derivatives in different solvents. **b** Stokes shifts of the anthracene derivatives as a function of the orientation polarizability calculated by Supplementary Equation (3). The solid lines are the fitting results by the Lippert-Mataga equation.

**Fig. 4** Phosphorescence. Phosphorescence spectra of the anthracene derivatives (2 wt%) dispersed in a polymer matrix. Spectra were integrated between 10 and 16 ms after 360 nm excitation at room temperature. Spectra are offset for clarity.

shows the energy diagram of the anthracene derivatives determined from the fluorescence and phosphorescence spectra along with twice the $T_1$ energies. These energy levels indicate that the TTU process is possible, because twice the $T_1$ energy (~3.5 eV) is higher than the $S_1$ energies. Figure 6 displays the energy diagrams of the anthracene derivatives determined from TD-DFT calculations (B3LYP/6-31G(d)). The $S_1$ and $T_1$ energies estimated

from the PL peak wavelengths agree well with the excitation energies obtained from the TD-DFT calculations. Assuming that the energy of $^3(TT)$ is slightly lower than twice the $T_1$ energy, as suggested by Kollmar et al.[28], the $T_2$ states obtained from the TD-DFT calculations are lower than $^3(TT)$. Therefore, the maximum $\eta_{TTU}$ of these anthracene derivatives should be limited to 20% according to the general TTU mechanism described above.

**Organic light-emitting diodes**. To investigate the relationship between CT characteristics and TTU, we fabricated OLEDs with a structure of indium tin oxide (ITO) (100 nm)/α-NPD (20 nm)/TAPC (20 nm)/anthracene derivative (20 nm)/TPBi (40 nm)/LiF (1 nm)/Al (80 nm), where α-NPD is $N,N'$-di-1-naphthyl-$N,N'$-diphenylbenzidine, TAPC is 1,1-bis[4-[$N,N$-di(p-tolyl)amino]phenyl]cyclohexane, and TPBi is 1,3,5-tris(1-phenyl-1$H$-benzimidazol-2-yl)benzene. Supplementary Figure 2 shows EL spectra of OLEDs with the anthracene derivatives as emitters at current densities ($J$) of 1, 10, 100, and 200 mA/cm² compared to PL spectra of these anthracene derivatives in thin films. The EL spectra are independent of $J$ and agree well with the PL spectra except for the DPA- and ANT-TRZ-based OLEDs. The difference of EL and PL spectra for the DPA- and ANT-TRZ-based OLEDs suggests the presence of small amounts of other excited-state species like exciplexes or electromers. However, we expect that these excited-state species do not affect the TTU process, which is caused by collisions between $^3$LE states. Supplementary Figures 3 and 4 show the $J$–voltage ($V$) characteristics and $\eta_{EQE}$–$J$ characteristics of OLEDs containing the anthracene derivatives. No

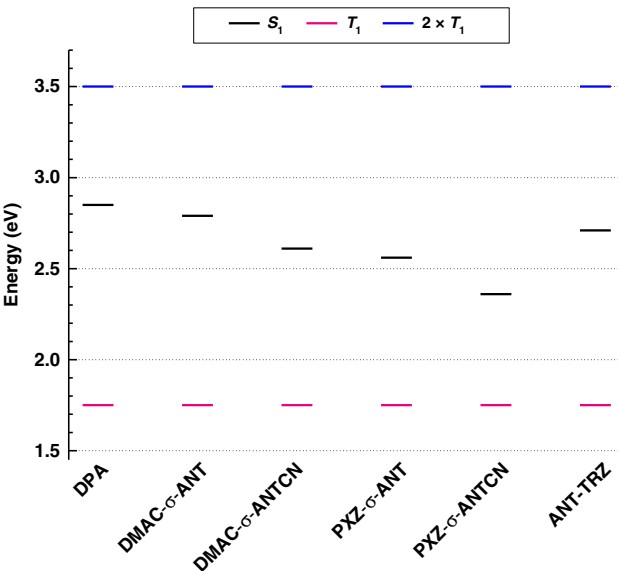

**Fig. 5** Excited-state energies. Energy levels of the anthracene derivatives determined from the fluorescence and phosphorescence spectra compared with twice the $T_1$ energies.

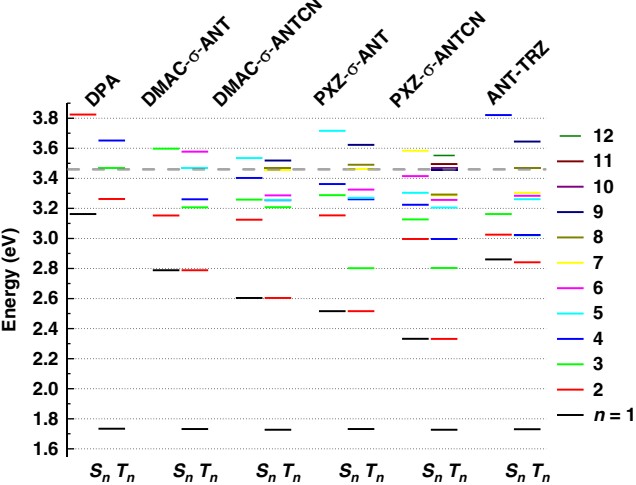

**Fig. 6** Energy diagrams. Energy levels and twice the $T_1$ energies (dashed line) of the anthracene derivatives calculated using B3LYP/6-31G(d), where $n$ is the quantum number used to indicate the excited state.

large differences between the carrier-transport properties of these anthracene derivatives were observed because their $J$–$V$ characteristics were similar. The EL data are listed in Table 1. To estimate $\eta_r$ independent of the carrier balance, we analysed transient EL characteristics. Figure 7a shows the transient EL characteristics of a PXZ-σ-ANTCN-based OLED at a current density ($J$) of 200 mA/cm$^2$ after the application of a pulse voltage for 25 ms. We observed prompt emission decay immediately after voltage cutoff and successive delayed decay emission. The observed delayed fluorescence $I(t)$ was fitted by a TTU model[14,19]:

$$I(t) = \frac{A}{1 + Bt}, \qquad (3)$$

where $A$ and $B$ are constants. Another possibility is that trapped carrier recombination is responsible for the delayed fluorescence. Delayed fluorescence originating from trapped carrier recombination should follow a power law with the scaling exponent between 1 and 2 (refs. [29,30]). Figure 7a shows the fitting results using these two models. The delayed fluorescence curve is well fitted by the TTU model. For other OLEDs, the observed delayed fluorescence curves are also well fitted by the TTU model, indicating that the delayed fluorescence observed from all OLEDs originates from TTU.

In these OLEDs, carrier recombination generates $^1$CT with a probability of 25% and $^3$CT with a probability of 75%. The latter relaxes to $T_1$, leading to the delayed fluorescence through TTU. In the transient EL characteristics, the prompt decays observed immediately after voltage cutoff are assigned to the EL intensities generated by direct recombination, $I_{EL}$(Prompt). In contrast, the delayed emission is assigned to the contribution of EL generated by TTU, $I_{EL}$(Delayed). Therefore, it is possible to experimentally evaluate the ratio of $I_{EL}$(Prompt) to $I_{EL}$(Delayed) and thus the contributions of TTU. At high triplet concentrations, the ratio of $I_{EL}$(Prompt) to $I_{EL}$(Delayed) is given by Eq. (21) in Methods. Using Eq. (21), we estimated $\eta_r$ in the OLEDs. Supplementary Figure 5 shows the dependence of $\eta_r$ estimated from the transient EL characteristics at $J = 1$ mA/cm$^2$ on pulse width. $\eta_r$ is saturated when the pulse width exceeds 0.1 ms, indicating the steady state

of the exciton density after a voltage is applied for 0.1 ms. Figure 7b presents $\eta_r$ of all the OLEDs estimated from the transient EL characteristics as a function of $J$. The estimated $\eta_r$ values of these OLEDs can be classified into two groups. The OLEDs containing DPA, DMAC-σ-ANT, and PXZ-σ-ANT show $\eta_r$ close to 40%. These efficiencies are within the limit of the general TTU model. Conversely, the estimated $\eta_r$ values of OLEDs with DMAC-σ-ANTCN, PXZ-σ-ANTCN, and ANT-TRZ exceeded the theoretical limit of the general TTU model. It is difficult to explain such superior TTU efficiency by the mechanism that applies to rubrene because several $T_n$ states are located between the $^3(TT)$ state and $T_1$ (Fig. 6). The observed maximum $\eta_{EQE}$ is lower than the theoretical EL efficiencies based on the estimated $\eta_r$ values. This suggests that the carrier balance factor is lower in these OLEDs.

To verify the advantage of TTU to improve device efficiency, we optimized the OLED structures for DMAC-σ-ANTCN and ANT-TRZ. The optimized device structures are ITO (100 nm)/ HAT-CN (10 nm)/Tris-PCz (30 nm)/DMAC-σ-ANTCN or 25 wt %-DMAC-σ-ANTCN: CBP (20 nm)/TmPyPB (40 nm)/LiF (1 nm)/Al (80 nm) and ITO (100 nm)/HAT-CN (10 nm)/Tris-PCz (30 nm)/ANT-TRZ or 25 wt%-ANT-TRZ: CBP (20 nm)/TPBi (40 nm)/LiF (1 nm)/Al (80 nm), where HAT-CN is hexaazatri-phenylenehexacarbonitrile, Tris-PCz is 9,9′-diphenyl-6-(9-phe-nyl-9$H$-carbazol-3-yl)-9$H$,9′$H$-3,3′-bicarbazole, TmPyPB is 1,3,5-tri[(3-pyridyl)-phen-3-yl]benzene, and CBP is 4,4′-Bis($N$-carba-zolyl)-1,1′-biphenyl. Figure 8 presents the $\eta_{EQE}$–$J$ characteristics of the optimized OLEDs using DMAC-σ-ANTCN and ANT-TRZ as emitting layers. The EL data are also listed in Table 1. Assuming $\eta_{OC}$ of 20% and $\eta_r$ of 62.5%, the observed maximum $\eta_{EQE}$ of 4.5% and 3.3% for the OLEDs with DMAC-σ-ANTCN and ANT-TRZ, respectively, are close to the theoretical EL efficiencies of 5.3% and 2.8%, respectively. On the other hand, assuming $\eta_{OC}$ of 20% and $\eta_r$ of 25%, the observed maximum $\eta_{EQE}$ of 3.2% and 1.9% for the OLEDs with DMAC-σ-ANTCN and ANT-TRZ doped CBP, respectively, are close to the theoretical EL efficiencies of 3.2% and 1.1%, respectively. With an increase of current density, the non-dope devices show efficiency roll-up due to TTU processes. On the other hand, the dope devices show no efficiency roll-up, indicating no TTA processes take place due to isolated molecules. These results would support TTA mechanisms in the non-dope OLEDs.

**Table 1 Photoluminescence data and electroluminescence data of the devices.**

| Emitting layer (EML) | PL $\lambda_{MAX}$[a] [nm] | $\Phi_{PL}$(film) [b] [%] | Device structure[c] | EL $\lambda_{MAX}$[d] [nm] | CIE (x, y)[e] $PE_{MAX}$[h] | $EQE_{MAX}$[f] [%] | $CE_{MAX}$[g] [cd/A] | $PE_{MAX}$[h] [lm/W] |
|---|---|---|---|---|---|---|---|---|
| DPA | 434 | 59.3 | A | 439 | 0.15, 0.08 | 2.0 | 1.4 | 1.3 |
| DMAC-σ-ANT | 446 | 42.8 | A | 446 | 0.16, 0.11 | 2.9 | 2.9 | 2.9 |
| DMAC-σ-ANTCN | 472 | 42.5 | A | 474 | 0.15, 0.23 | 2.5 | 4.3 | 4.2 |
| PXZ-σ-ANT | 488 | 19.1 | A | 487 | 0.17, 0.36 | 2.0 | 4.4 | 4.1 |
| PXZ-σ-ANTCN | 526 | 15.5 | A | 520 | 0.29, 0.56 | 2.0 | 6.2 | 4.4 |
| ANT-TRZ | 456 | 22.6 | A | 465 | 0.15, 0.20 | 0.9 | 1.3 | 0.7 |
| DMAC-σ-ANTCN | 472 | 42.5 | B | 478 | 0.15, 0.24 | 4.5 | 8.2 | 4.1 |
| 25 wt%-DMAC-σ-ANTCN:CBP | 455 | 64.3 | B | 467 | 0.15, 0.16 | 3.2 | 4.2 | 3.9 |
| ANT-TRZ | 456 | 22.6 | C | 465 | 0.15, 0.19 | 3.3 | 4.9 | 2.6 |
| 25 wt%-ANT-TRZ:CBP | 454 | 21.5 | C | 466 | 0.15, 0.17 | 1.9 | 2.5 | 1.4 |

[a]The maximum electroluminescence wavelength with the excitation wavelength of 360 nm. [b]The photoluminescence quantum yield of the thin film under argon atmosphere with the excitation wavelength of 360 nm. [c]The device structures are: (A) ITO (100 nm)/α-NPD (20 nm)/TAPC (20 nm)/EML (20 nm)/TPBi (40 nm)/LiF (1 nm)/Al (80 nm), (B) ITO (100 nm)/HAT-CN (10 nm)/Tris-PCz (30 nm)/EML (20 nm)/TmPyPB (40 nm)/LiF (1 nm)/Al (80 nm), and (C) ITO (100 nm)/HAT-CN (10 nm)/Tris-PCz (30 nm)/EML (20 nm)/TPBi (40 nm)/LiF (1 nm)/Al (80 nm), respectively. [d]The maximum electroluminescence wavelength. [e]Commission internationale de l'Eclairage (CIE) recorded at 200 mA/cm². [f]The maximum external electroluminescence efficiency. [g]The maximum current efficiency [h]The maximum power efficiency

**TTU mechanism.** We now discuss a possible mechanism for the maximum TTU efficiencies of some of the anthracene derivatives exceeding the limit of the general TTU model. By assuming that $^3$(TT) is dissociated into two $T_1$ or converted to $^1$(TT), the TTU efficiency should be improved. However, these scenarios cannot explain the dependence of the maximum TTU efficiency on the material because their $T_1$ states are assigned to $^3$LE of the anthracene unit, so it is expected that $^3$(TT) of these derivatives are almost identical. Therefore, we focus on the possibility of a conversion process from $^3$(TT) to $S_1$.

Immediately after the TTU transitions, the higher singlet excited state and higher triplet excited state would form and then relax to $S_1$ and $T_1$, respectively, via internal conversion. TTU transitions are the reverse process of singlet fission (SF) transitions. The rate constant of SF is given by Fermi's golden rule involving the electron–electron interaction ($H_{el}$)[31]. To derive the rate from the triplet intermediate state ($^3$(TT)) to $S_n$, we consider the triplet intermediate state ($|^3T_1T_1\rangle$) mixing $|T_mS_0\rangle$, which represents a pair of molecules in $T_n$ and $S_0$, involving the perturbation of $H_{el}$ similar to the SF process.

$$\left|^3T_1T_1\right\rangle = \left|^3T_1T_1^{(0)}\right\rangle + \sum_{m=1} \frac{\left\langle T_mS_0|H_{el}|^3T_1T_1^{(0)}\right\rangle}{E\left(^3T_1T_1^{(0)}\right) - E(T_mS_0)}|T_mS_0\rangle,$$

(4)

where $S_n$ and $T_m$ are the $n$th singlet excited state and the $m$th triplet excited state, respectively; $\left|^3T_1T_1^{(0)}\right\rangle$ is the unperturbed $^3$(TT); and $E\left(^3T_1T_1^{(0)}\right)$ and $E(T_mS_0)$ are the energies of $\left|^3T_1T_1^{(0)}\right\rangle$ and $|T_mS_0\rangle$, respectively. Based on the perturbed state, the rate from $^3$(TT) to $S_n$ ($k_1$) is given by Fermi's golden rule involving the spin–orbit interaction ($H_{so}$),

$$k_1 = \frac{2\pi}{\hbar}\left|\frac{\langle S_nS_0|H_{so}|T_mS_0\rangle\left\langle T_mS_0|H_{el}|^3T_1T_1^{(0)}\right\rangle}{E\left(^3T_1T_1^{(0)}\right) - E(T_mS_0)}\right|^2 \delta\left(E(S_nS_0) - E\left(^3T_1T_1^{(0)}\right)\right),$$

(5)

where the $|S_nS_0\rangle$ state represents a pair of molecules in $S_n$ and $S_0$ states and $E(S_nS_0)$ is the energy of $|S_nS_0\rangle$. Here, the term $\langle S_nS_0|H_{so}|^3T_1T_1^{(0)}\rangle$ is negligible. As a result, $k_1$ can be expressed as

follows:

$$k_1 = \frac{2\pi}{\hbar}\left|\frac{\langle S_n|H_{so}|T_m\rangle}{E\left(^3T_1T_1^{(0)}\right) - E(T_mS_0)}\right|^2 \left|\left\langle T_mS_0|H_{el}|^3T_1T_1^{(0)}\right\rangle\right|^2$$
$$\delta\left(E(S_nS_0) - E\left(^3T_1T_1^{(0)}\right)\right).$$

(6)

The first integral is the one-electron integral related to the spin–orbit matrix element and the second integral is the matrix element of the perturbation of $H_{el}$ related to the transition from $|^3T_1T_1\rangle$ to $|T_mS_0\rangle$. The second integral should be enough for this transition because the transition from $|^3T_1T_1\rangle$ to $|T_mS_0\rangle$ can occur. Therefore, the conversion process from $^3$(TT) to $S_n$ can contribute to the TTU efficiency when the spin–orbit matrix element is large compared to the energy difference between $|^3T_1T_1\rangle$ and $|T_mS_0\rangle$. Here, we emphasize that this spin conversion process from $|^3T_1T_1\rangle$ to $|S_nS_0\rangle$ takes place directly without going through the transition to the $|T_mS_0\rangle$, similar to a transition between $S_0$ and $T_1$, mixing $S_1$ as the perturbation terms[32]. Therefore, there is no internal conversion from $|T_mS_0\rangle$ and $|S_1S_0\rangle$. The conversion process from $^3$(TT) to $S_n$ is strongly affected by the spin–orbit matrix element. To enhance the spin–orbit interaction without a heavy-metal effect, a transition between molecular orbitals with magnetic moments in different directions should be involved; that is, the so-called El-Sayed rule should apply[33–35]. For example, the transition between $^1$CT and $^3$LE often enhances the spin–orbit interaction.

According to Eq. (6), $T_m$ should be close to the energy of $^3$(TT) to facilitate $k_1$. Supplementary Table 3 gives the electronic transitions related to $T_m$ with energies in the range of 3.45–3.50 eV. Immediately after the TTU transitions, $S_n$ would form, which then relaxes to $S_1$ via internal conversion. Therefore, we considered the spin–orbit interaction between $S_n$ with an energy lower than twice the $T_1$ energy (~3.5 eV) and the selected $T_m$. Supplementary Table 3 also lists the electronic transitions related to the considered $S_n$. The spin–orbit matrix elements related to the possible TTU transitions from $^3$(TT) to $S_n$ are summarized in Table 2. In DMAC-σ-ANT, the TTU transition from $^3$(TT) to $S_n$ involves the spin–orbit matrices consisting of the molecular orbitals of the highest occupied molecular orbital (HOMO), HOMO-1, and HOMO-4 (Table 2).

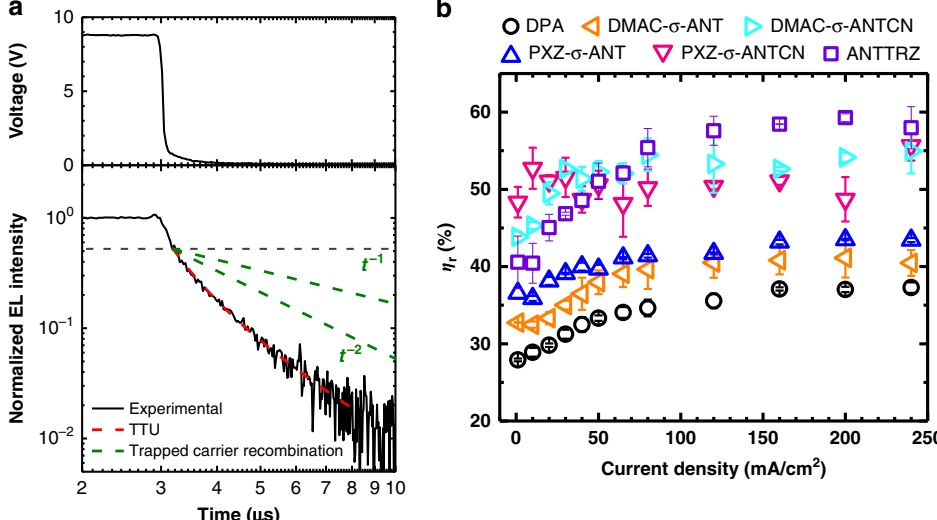

**Fig. 7** TTU efficiency in OLEDs. **a** Transient voltage (top) and EL intensity (bottom, black solid line) characteristics and the fitting results using the TTU model (bottom, red dashed line) and trapped carrier recombination model (bottom, green dashed line). **b** Dependence of the radiative-exciton production efficiency ($\eta_r$) on current density ($J$) estimated from transient EL characteristics.

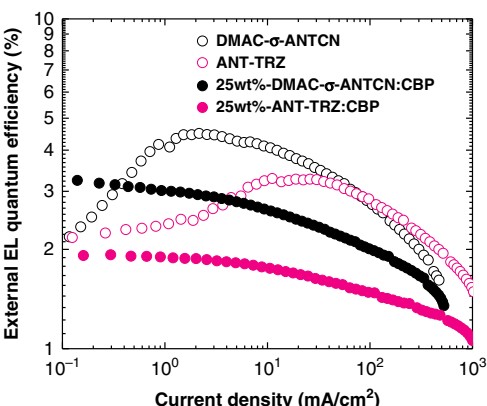

**Fig. 8** Performance of the TTU-OLEDs. External EL quantum efficiency ($\eta_{EQE}$) as a function of current density ($J$) for devices with structures of ITO (100 nm)/HAT-CN (10 nm)/Tris-PCz (30 nm)/DMAC-σ-ANTCN or 25wt%- DMAC-σ-ANTCN: CBP (20 nm)/TmPyPB (40 nm)/LiF (1 nm)/Al (80 nm) (black open circles and filled circles) and ITO (100 nm)/HAT-CN (10 nm)/Tris-PCz (30 nm)/ANT-TRZ or 25wt%-ANT-TRZ: CBP (20 nm)/ TPBi (40 nm)/LiF (1 nm)/Al (80 nm) (pink open circles and filled circles).

As shown in Supplementary Fig. 7, these orbitals are localized on dimethylacridine or anthracene units, which have molecular surfaces in the same direction and magnetic moments oriented in the direction vertical to these surfaces. Therefore, we expect small spin–orbit matrix elements because different vectors of the magnetic moments are not involved in the TTU process. Conversely, for DMAC-σ-ANTCN (Supplementary Fig. 8), PXZ-σ-ANTCN (Supplementary Fig. 10), and ANT-TRZ (Supplementary Fig. 11), the different vectors of the magnetic moments are involved in the TTU process. For example, in ANT-TRZ, as presented in Table 2 and Supplementary Fig. 11, the TTU transition from $^3(TT)$ to $S_n$ involves the spin–orbit matrix of $\langle S_1|H_{so}|T_8\rangle$ consisting of the lowest unoccupied molecular orbital (LUMO) and LUMO + 4. The LUMO is localized on the 2,4,6-triphenyl-1,3,5-triazine unit, whereas the

LUMO + 4 is mainly localized on the anthracene unit, which is oriented perpendicular to the triazine unit. As a result, the spin–orbit matrices involve the rotation of the molecular orbitals from the anthracene unit to the triazine unit. Therefore, it is possible that the spin–orbit matrix element would be increased according to the El-Sayed rule, leading to TTU involving spin conversion. In DPA (Supplementary Fig. 6) and PXZ-σ-ANT (Supplementary Fig. 9), we also identified the possibility of the spin–orbit interaction being increased according to the El-Sayed rule. Conversely, we observed no contribution from the spin conversion process from $^3(TT)$ to $S_1$ in the OLEDs with DPA and PXZ-σ-ANT. Although unclarified factors may impede the upconversion, transitions involving molecular orbitals with relative orthogonal directions would be a probable mechanism for the conversion process from $^3(TT)$ to $S_1$.

Our results are very different from the RISC model arising from "hot" CT channels because we observed delayed fluorescence in all OLEDs that originated from TTU[26,36,37]. However, the mechanism of spin conversion proposed to occur in selected anthracene derivatives would be similar to the "hot" CT channels, which are derived from second-order perturbations including nonadiabatic and spin–orbit coupling terms[33].

## Discussion

We studied the TTU efficiency of anthracene derivatives with different electron-donating and -withdrawing substituents by including them as emitters in OLEDs. In the OLEDs using DMAC-σ-ANTCN, PXZ-σ-ANTCN, and ANT-TRZ, the TTU efficiency exceeded the theoretical limit of the conventional TTU model. The results suggested the conversion process from $^3(TT)$ to $S_1$ occurred in these OLEDs. To understand these results, we proposed a TTU model involving a spin–orbit interaction based on Fermi's golden rule. Compared with the superior TTU limited to a few materials[13,15,19,24] and the "hot" CT channels through HLCT[26,36,37], the advantage of the proposed molecular designs can escape the tight restriction of the $T_2$ energies. Therefore, this work should pave the way to realize efficient deep-blue and ultraviolet OLEDs.

**Table 2 Spin–orbit matrices related to possible TTU transitions from $^3$(TT) to $S_n$ for the anthracene derivatives**

| Compounds | Transition | SOC matrix | | El'Sayed's rule | Related molecular orbitals |
|---|---|---|---|---|---|
| DPA | $^3T_1T_1 \rightarrow S_1$ | $\langle S_1|H_{so}|T_3\rangle$ | $\langle HOMO|H_{so}|HOMO\text{-}1\rangle$ | Unsatisfied | Supplementary Fig. 6 |
| | | | $\langle LUMO|H_{so}|LUMO\text{+}1\rangle$ | Satisfied | |
| DMAC-σ-ANT | $^3T_1T_1 \rightarrow S_1$ | $\langle S_1|H_{so}|T_5\rangle$ | $\langle HOMO|H_{so}|HOMO\text{-}4\rangle$ | Unsatisfied | Supplementary Fig. 7 |
| | $^3T_1T_1 \rightarrow S_2$ | $\langle S_2|H_{so}|T_5\rangle$ | $\langle HOMO\text{-}1|H_{so}|HOMO\text{-}4\rangle$ | Unsatisfied | |
| DMAC-σ-ANTCN | $^3T_1T_1 \rightarrow S_1$ | $\langle S_1|H_{so}|T_7\rangle$ | $\langle HOMO|H_{so}|HOMO\text{-}5\rangle$ | Unsatisfied | Supplementary Fig. 8 |
| | $^3T_1T_1 \rightarrow S_2$ | $\langle S_2|H_{so}|T_7\rangle$ | $\langle HOMO\text{-}1|H_{so}|HOMO\text{-}5\rangle$ | Unsatisfied | |
| | | $\langle S_2|H_{so}|T_8\rangle$ | $\langle LUMO|H_{so}|LUMO\text{+}1\rangle$ | Satisfied | |
| | $^3T_1T_1 \rightarrow S_3$ | $\langle S_3|H_{so}|T_8\rangle$ | $\langle HOMO|H_{so}|HOMO\text{-}8\rangle$ | Satisfied | |
| | | | $\langle HOMO|H_{so}|HOMO\text{-}1\rangle$ | Unsatisfied | |
| | $^3T_1T_1 \rightarrow S_4$ | $\langle S_4|H_{so}|T_8\rangle$ | $\langle HOMO\text{-}1|H_{so}|HOMO\text{-}8\rangle$ | Satisfied | |
| PXZ-σ-ANT | $^3T_1T_1 \rightarrow S_1$ | $\langle S_1|H_{so}|T_7\rangle$ | $\langle HOMO|H_{so}|HOMO\text{-}3\rangle$ | Unsatisfied | Supplementary Fig. 9 |
| | | $\langle S_1|H_{so}|T_8\rangle$ | $\langle LUMO|H_{so}|LUMO\text{+}8\rangle$ | Unsatisfied | |
| | $^3T_1T_1 \rightarrow S_2$ | $\langle S_2|H_{so}|T_7\rangle$ | $\langle HOMO\text{-}1|H_{so}|HOMO\text{-}3\rangle$ | Unsatisfied | |
| | $^3T_1T_1 \rightarrow S_3$ | $\langle S_3|H_{so}|T_8\rangle$ | $\langle LUMO\text{+}2|H_{so}|LUMO\text{+}8\rangle$ | Satisfied | |
| | $^3T_1T_1 \rightarrow S_4$ | $\langle S_4|H_{so}|T_8\rangle$ | $\langle LUMO\text{+}1|H_{so}|LUMO\text{+}8\rangle$ | Satisfied | |
| PXZ-σ-ANTCN | $^3T_1T_1 \rightarrow S_1$ | $\langle S_1|H_{so}|T_9\rangle$ | $\langle HOMO|H_{so}|HOMO\text{-}3\rangle$ | Unsatisfied | Supplementary Fig. 10 |
| | | $\langle S_1|H_{so}|T_{11}\rangle$ | $\langle LUMO|H_{so}|LUMO\text{+}9\rangle$ | Unsatisfied | |
| | $^3T_1T_1 \rightarrow S_2$ | $\langle S_2|H_{so}|T_{10}\rangle$ | $\langle HOMO|H_{so}|HOMO\text{-}8\rangle$ | Satisfied | |
| | | | $\langle HOMO|H_{so}|HOMO\text{-}1\rangle$ | Unsatisfied | |
| | | $\langle S_2|H_{so}|T_{11}\rangle$ | $\langle LUMO\text{+}1|H_{so}|LUMO\text{+}9\rangle$ | Satisfied | |
| | $^3T_1T_1 \rightarrow S_3$ | $\langle S_3|H_{so}|T_9\rangle$ | $\langle HOMO\text{-}1|H_{so}|HOMO\text{-}3\rangle$ | Unsatisfied | |
| | | $\langle S_3|H_{so}|T_{10}\rangle$ | $\langle LUMO|H_{so}|LUMO\text{+}1\rangle$ | Satisfied | |
| | $^3T_1T_1 \rightarrow S_4$ | $\langle S_4|H_{so}|T_{11}\rangle$ | $\langle LUMO\text{+}4|H_{so}|LUMO\text{+}9\rangle$ | Satisfied | |
| | $^3T_1T_1 \rightarrow S_5$ | $\langle S_5|H_{so}|T_{11}\rangle$ | $\langle LUMO\text{+}3|H_{so}|LUMO\text{+}9\rangle$ | Satisfied | |
| | $^3T_1T_1 \rightarrow S_6$ | $\langle S_6|H_{so}|T_{10}\rangle$ | $\langle HOMO\text{-}1|H_{so}|HOMO\text{-}8\rangle$ | Satisfied | |
| ANT-TRZ | $^3T_1T_1 \rightarrow S_1$ | $\langle S_1|H_{so}|T_8\rangle$ | $\langle LUMO|H_{so}|LUMO\text{+}4\rangle$ | Satisfied | Supplementary Fig. 11 |
| | $^3T_1T_1 \rightarrow S_2$ | $\langle S_2|H_{so}|T_8\rangle$ | $\langle LUMO\text{+}1|H_{so}|LUMO\text{+}4\rangle$ | Satisfied | |
| | $^3T_1T_1 \rightarrow S_3$ | $\langle S_3|H_{so}|T_8\rangle$ | $\langle HOMO|H_{so}|HOMO\text{-}1\rangle$ | Unsatisfied | |
| | | | $\langle LUMO\text{+}2|H_{so}|LUMO\text{+}4\rangle$ | Satisfied | |

## Methods

**Measurements**. $^1$H and $^{13}$C NMR spectra were obtained in CDCl$_3$ with a Bruker Biospin Avance-III 500 NMR (Germany) spectrometer at ambient temperature. Absorption, PL, and phosphorescence spectra were measured using a UV–Vis spectrometer (Lambda 950; Perkin-Elmer, United states), a spectrofluorometer (FluoroMax-3, Horiba, Japan), and another spectrofluorometer (FP-8600, JASCO, Japan), respectively. Absolute PL quantum yields were measured using a Quantaurus-QY absolute PL quantum yield spectrometer (C11347-11; Hamamatsu Photonics, Japan) under 360-nm excitation and an Ar flow. The thicknesses of the materials were measured by variable angle spectroscopic ellipsometry (M-2000U; J. A. Woollam Co., Inc., United states). Transient PL decay curves were measured using a Quantaurus-Tau fluorescence lifetime measurement system (C11367-03; Hamamatsu Photonics, Japan).

EL spectra, $J–V$ characteristics, and $\eta_{EQE}–J$ characteristics were measured using an absolute EQE measurement system (C9920-12, Hamamatsu Photonics, Japan). Transient EL characteristics were measured using a photomultiplier tube (R928; Hamamatsu Photonics, Japan) connected to an amplifier unit (C6438; Hamamatsu Photonics, Japan) under pulsed driving using a pulse generator (8114A; Agilent, United states). Signals were monitored using an oscilloscope (TBS2104, Tektoronix, United states).

**Film and OLED fabrication**. Thin films for optical measurements were fabricated by vacuum vapour deposition under a pressure of less than $10^{-3}$;Pa on quartz substrates. A fixed deposition rate of 0.3 nm/s and thickness of 100 nm were used. OLEDs were fabricated on ITO-coated glass substrates by vacuum vapour deposition under a pressure of less than $5.0 \times 10^{-4}$ Pa without atmospheric exposure. Organic layers were deposited at a deposition rate of 0.3 nm/s through a metal mask. After deposition, the metal mask was replaced with another metal mask for cathode deposition in a nitrogen-filled glove box. The electrodes were deposited on the organic layers at deposition rates of 0.02 nm/s for LiF and 0.05–0.5 nm/s for Al.

**Solvent effects of the anthracene derivates**. The solvent effects on Stokes shifts are modelled by the Lippert–Mataga equation as follows[26]:

$$\tilde{\nu}_a - \tilde{\nu}_f = \frac{2\left(\mu_e - \mu_g\right)^2}{hca^3}f(\varepsilon, n) + \text{constant}, \tag{7}$$

$$\frac{4\pi a^3}{3} = \frac{M}{N_A d}, \tag{8}$$

where $\tilde{\nu}_a(cm^{-1})$ is the peak wave-number of absorption at the low-frequency side; $\tilde{\nu}_f(cm^{-1})$ is the peak wave-number of fluorescence at the high-frequency side; $\mu_e$ and $\mu_g$ are the dipole moments of the excited state and the ground state, respectively; $h$ (erg) is the Planck constant; $c$ is the speed of light (cm/s); $a$ is the molecular radius (cm); $M$ is the molecular weight; $N_A$ is the Avogadro number; $d$ is the density of solid states assuming 1.0 g/cm$^3$; $f(\varepsilon, n)$ is the orientation polarizability given by

$$f(\varepsilon, n) = \frac{\varepsilon - 1}{2\varepsilon + 1} - \frac{n^2 - 1}{2n^2 + 1}, \tag{9}$$

where $\varepsilon$ and $n$ are the dielectric constant and the refractive index of the solvent, respectively. The $\mu_g$ is calculated by density function theory (DFT) calculations (B3LYP/6-31G(d)). The calculated $\mu_g$ of DPA, DMAC-σ-ANT, DMAC-σ-ANTCN, PXZ-σ-ANT, PXZ-σ-ANTCN, and ANT-TRZ are 0.0, 1.8, 3.8, 3.0, 2.5, and 0.2 D, respectively. The data used from the Lippert–Mataga plots are listed in Supplementary Table 2.

**Measurement of the room-temperature phosphorescence of the anthracene derivatives**. Phosphorescence spectra of the anthracene derivatives were observed at room temperature using a reported method[27]. To enhance phosphorescence from the anthracene derivatives, we used a composite host consisting of poly(4-bromostyrene) and benzophenone with a weight ratio of 5:1. The anthracene derivatives (2 wt%) were dispersed in the host matrix. These materials were dissolved in methoxybenzene to give a total weight concentration of 240 mg/mL. Samples were fabricated by drop casting the methoxybenzene solution on quartz substrates. After drop casting, the samples were dried in the ambient atmosphere.

**The ratio of EL intensity generated by direct recombination and TTU**. Assuming that the singlet and triplet intermediate states formed by triplet collision are completely converted into $S_1$ and $T_1$ without returning to two $T_1$, the respective rate equations of the singlet and triplet densities ($S$ and $T$) are given by Eqs (10) and (11), respectively.

$$\frac{dS}{dt} = \frac{1}{4}G - \left(k_r^S + k_{nr}^S + k_{ISC}\right)S + \frac{\alpha_S}{2}\gamma_{TT}T^2, \tag{10}$$

$$\frac{dT}{dt} = \frac{3}{4}G - k_{nr}^T T - \gamma_{TT} T^2 + k_{ISC} S + \frac{\alpha_T}{2}\gamma_{TT} T^2 + \alpha_Q \gamma_{TT} T^2, \qquad (11)$$

where $G$ is the recombination rate constant; $k_r^S$ and $k_{nr}^S$ are the radiative and nonradiative rate constants of the singlet state, respectively; $k_{ISC}$ is the rate constant of intersystem crossing (ISC) from singlet to triplet states; $k_{nr}^T$ is the nonradiative rate constant of the triplet state; $\gamma_{TT}$ is the TTU rate constant; and $\alpha_S$, $\alpha_T$, and $\alpha_Q$ are the formation probabilities of singlet, triplet, and quintet intermediate states, respectively. Here, we define the TTU rate constant excluding the quintet formation process, $\gamma_{TT}'$, and the relative conversion efficiency from two triplet states to one singlet state by a single TTU event, $\alpha$.

$$\gamma_{TT}' = (\alpha_S + \alpha_T)\gamma_{TT}, \qquad (12)$$

$$\alpha = \frac{\alpha_S}{\alpha_S + \alpha_T}, \qquad (13)$$

$$\alpha_S + \alpha_T + \alpha_Q = 1, \qquad (14)$$

Equations (10) and (11) can be simplified using Equation S6–S8 to give

$$\frac{dS}{dt} = \frac{1}{4}G - \left(k_r^S + k_{nr}^S + k_{ISC}\right)S + \frac{\alpha}{2}\gamma_{TT}' T^2, \qquad (15)$$

$$\frac{dT}{dt} = \frac{3}{4}G - k_{nr}^T T - \gamma_{TT}' T^2 + k_{ISC} S + \frac{1-\alpha}{2}\gamma_{TT}' T^2. \qquad (16)$$

In the general TTU model, $\alpha$ is limited to 25%. In these equations, the last terms represent the generation of singlet and triplet excitons by TTU. In the strong TTU limit, where $\gamma_{TT} T$ is much larger than $k_{nr}^T$, Eq. (16) can be simplified to

$$\frac{dT}{dt} = \frac{3}{4}G + k_{ISC} S - \frac{1+\alpha}{2}\gamma_{TT}' T^2. \qquad (17)$$

This condition would be achieved at high $J$. In the steady state of the exciton densities, the total EL intensity, $I$(Total), can be obtained from Eqs (15) and (17).[15]

$$I_{EL}(\text{Total}) = G \times \Phi_{PL} \times \eta_{OC} \times \frac{1+4\alpha}{4+4\alpha-4\Phi_{ISC}\alpha}, \qquad (18)$$

where $\Phi_{ISC}$ is the ISC yield. The last term represents the radiative-exciton production efficiency ($\eta_r$). Here, the total EL is divided into the contributions of EL generated by direct recombination and TTU. The EL intensity generated by direct recombination, $I_{EL}$(Prompt), is given by

$$I_{EL}(\text{Prompt}) = G \times \Phi_{PL} \times \eta_{OC} \times \frac{1}{4}. \qquad (19)$$

Thus, the EL intensity generated by TTU, $I_{EL}$(Delayed), can be obtained by subtracting Eq. (19) from Eq. (18)

$$I_{EL}(\text{Delayed}) = G \times \Phi_{PL} \times \eta_{OC} \times \frac{3\alpha + \Phi_{ISC}\alpha}{4+4\alpha-4\Phi_{ISC}\alpha}. \qquad (20)$$

As a result, the ratio of these contributions is given by

$$\frac{I_{EL}(\text{Delayed})}{I_{EL}(\text{Prompt})} = \frac{3\alpha + \Phi_{ISC}\alpha}{1+\alpha-\Phi_{ISC}\alpha} = 4\left(\frac{1}{4} + \frac{3\alpha + \Phi_{ISC}\alpha}{4+4\alpha-4\Phi_{ISC}\alpha}\right) - 1 = 4\eta_r - 1. \qquad (21)$$

## Data availability

The data that support the findings of this study are available from the corresponding author upon reasonable request.

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

## Acknowledgements
This work was supported by the Japan Science and Technology Agency (JST), ERATO, Adachi Molecular Exciton Engineering Project (Grant Number JPMJER1305), the International Institute for Carbon Neutral Energy Research (WPI-I$^2$CNER) sponsored by the Ministry of Education, Culture, Sports, Science and Technology (MEXT), MEXT/JSPS KAKENHI (Grant Number JP 16H06057), and Kyulux Inc.

## Author contributions
R.I. and K.G. designed and conducted the experiments, and discussed the experimental data. R.I., K.G., and C.A. wrote the manuscript.

## Competing interests
The authors declare no competing interests.
