## [Peer Review File · Nature Communications]

Reviewers' comments:

Reviewer #1 (Remarks to the Author):

In this manuscript, authors studied the influence of electron-donating and -withdrawing substituents on the TTU efficiency of a series of anthracene derivatives. They found that the TTU efficiency of some of these derivatives reached up to 50%, which greatly exceeded the theoretical limit of the conventional TTU model. Combined with the molecular structures analysis and quantum chemical calculations, the authors attributed the enhanced TTU efficiency to the conversion process from $3(TT)$ to $S1$. This research provides constructive insights into the further decoding of the underlying mechanism of triplet-triplet upconversion. And it is therefore recommended to be published in this journal after some minor revisions.

1. According to some early reports, it is usually difficult to directly measure the phosphorescent spectra of anthracene derivatives as the impeded ISC process due to the large energy gap between $S1$ and $T1$. In Figure S2, the authors got very appreciable phosphorescence spectra. The cited reference No.27 doesn't contain detailed information. It would be better to give more measurement details, such as the excitation wavelength and the molecular structure of polymer matrix.
2. Some key electroluminescence data needs to be provided in the manuscript, such as electroluminescence peaks, Commission Internationale de l'Éclairage (CIE) coordinates, the maximum brightness, the maximum current efficiency, etc.
3. Featureless PL spectra were obtained for DMAC- σ -ANTCN, PXZ- σ -ANT, PXZ- σ -ANTCN, and ANT-TRZ, indicating each emission originated from CT. However, some local exciton (LE) molecules also show featureless PL spectra. Please prove this statement by the solvatochromic effect.
4. The maximum η_{EQE} is observed to be 4.5% and 3.3% for the OLEDs based on DMAC- σ -ANTCN and ANT-TRZ, respectively, which is inferior to a series of recently reported anthracene derivatives (Adv. Mater. 2019, 1807388; Adv. Funct. Mater. 2018, 28, 1803369; J. Mater. Chem. C, 2019, 7, 1014-1021). Thereinto, OLED based on carbazol-benzonitrile-anthracene, which possesses the similar molecular structure with DMAC- σ -ANTCN and PXZ- σ -ANTCN, has achieved EQE over 10% in nondoped device (J. Mater. Chem. C, 2019, 7, 1014-1021). Is it possible for the authors to give some comments on this?
5. Some grammar errors should be modified in the manuscript. For instance, on page 9, line 9, "and thus the the contributions".
6. In reference No.17, the publication year is wrong. It should be Adv. Funct. Mater. 24, 2064-2071 (2014).

Reviewer #2 (Remarks to the Author):

In this manuscript, the authors developed a series of new anthracene derivatives to increase the triplet-triplet upconversion efficiency in OLEDs. With the help of quantum chemical calculations, the authors proposed a possible upconversion mechanism which involves spin-orbit interaction between S_n and T_m . OLED efficiencies of 4.5% and 3.3% were achieved by DMAC- σ -AnTCN and ANT-TRZ, respectively. But notable efficiency roll-off at higher current density was also observed. In this manuscript, the authors tried to prove that the conversion from $3(TT)$ to S_n improves the TTU efficiency, but the conclusion is not very convincing since there are several problems contained in this manuscript.

1. The ISC quantum yields of the anthracene derivatives in the films are not convincing enough for further calculation of α and η_r . The authors stated in the supporting information (page 34) that k_r and k_{ISC} were assumed to be the same for the solution sample and thin film, but I do not agree with the statement. The energy levels of excited states, especially CT states, are quite sensitive to medium properties. From the photophysical data in Table S3, we can learn that the PLQYs in films were much lower than the PLQYs in solutions, suggesting very different photophysical properties in

different medium. The authors need to conduct PL lifetime measurements as well as other characterization in the films to support the calculation.

2. The mechanism of the spin conversion process in this manuscript is very similar to the "hot" CT channels, but the authors need to clarify how the IC process was blocked in this system. For example, the author stated that the spin conversion process in ANT-TRZ was from T8 to S1 (page 13). But according to the calculated energy levels presented in Fig. S3, the energy gaps between T8 and other triplet excited states (T2 to T7) are relatively small. Therefore the internal conversion should be much faster than the spin conversion process. An explanation is needed.

In addition, the TTU model in the manuscript seems like a combination of TTA and HLCT process, both of which are efficient strategies in OLEDs to exceed the theoretical limit of conventional fluorescence materials. But the device efficiencies in this manuscript are rather low with notable efficiencies roll-off. I think the authors need to reorganize the manuscript and conclude the advantages of the design strategies compared with the reported examples (TTA and HLCT). Perhaps this will shed light on the OLED emitter design. With those changes through a major revision, this work may merit publication in the Nature Communications.

We would like to take this opportunity to thank the reviewers for his/her valuable comments. In response to the reviewers' comments, we have made the following changes and responses (All changes have been highlighted in yellow):

Reviewer #1 (Remarks to the Author):

In this manuscript, authors studied the influence of electron-donating and -withdrawing substituents on the TTU efficiency of a series of anthracene derivatives. They found that the TTU efficiency of some of these derivatives reached up to ~ 50%, which greatly exceeded the theoretical limit of the conventional TTU model. Combined with the molecular structures analysis and quantum chemical calculations, the authors attributed the enhanced TTU efficiency to the conversion process from $^3(\text{TT})$ to S_1 . This research provides constructive insights into the further decoding of the underlying mechanism of triplet–triplet upconversion. And it is therefore recommended to be published in this journal after some minor revisions.

Q1. According to some early reports, it is usually difficult to directly measure the phosphorescent spectra of anthracene derivatives as the impeded ISC process due to the large energy gap between S_1 and T_1 . In Figure S2, the authors got very appreciable phosphorescence spectra. The cited reference No.27 doesn't contain detailed information. It would be better to give more measurement details, such as the excitation wavelength and the molecular structure of polymer matrix.

A1. We made mistake for the cited reference. The reference should be No. #27 in the revised manuscript. We are sorry for the mistake. In the revised Supplementary Information, we also added to the experimental details.

Q2. Some key electroluminescence data needs to be provided in the manuscript, such as electroluminescence peaks, Commission Internationale de l'Éclairage (CIE) coordinates, the maximum brightness, the maximum current efficiency, etc.

A2. Thank you for this comment. We summarized the electroluminescence data in Table S3.

Q3. Featureless PL spectra were obtained for DMAC- σ -ANTCN, PXZ- σ -ANT, PXZ- σ -ANTCN, and ANT-TRZ, indicating each emission originated from CT. However, some local exciton (LE) molecules also show featureless PL spectra. Please prove this statement by the solvatochromic effect.

A3. Thank you for this valuable comment. We measured the solvent dependence of absorption and PL spectra for the anthracene derivatives. In order to assign the excited state to LE, CT or the mixing, we conducted Lippert-Mataga plot as shown in Fig. S2b. Although PXZ- σ -ANT, PXZ- σ -ANTCN, and ANT-TRZ show the monotonical slope indicating that their excited states originate from a CT state, while DMAC- σ -ANT and DMAC- σ -ANTCN show two independent slopes,

suggesting the mixing of LE and CT states. The PL spectrum of the DMAC- σ -ANTCN film is close to the PL spectra in the polar solvents where the CT emission is dominant, while we can recognize the weak shoulder peaks at the shorter wavelength region in the broad spectra. Therefore, the singlet excited state of DMAC- σ -ANTCN in the films would be assigned to a mixed state of 1 CT weakly coupled with 1 LE. Based on these experimental data, we revised the description.

Q4. The maximum η_{EQE} is observed to be 4.5% and 3.3% for the OLEDs based on DMAC- σ -ANTCN and ANT-TRZ, respectively, which is inferior to a series of recently reported anthracene derivatives (Adv. Mater. 2019, 1807388; Adv. Funct. Mater. 2018, 28, 1803369; J. Mater. Chem. C, 2019, 7, 1014-1021). Thereinto, OLED based on carbazole-benzonitrile-anthracene, which possesses the similar molecular structure with DMAC- σ -ANTCN and PXZ- σ -ANTCN, has achieved EQE over 10% in nondoped device (J. Mater. Chem. C, 2019, 7, 1014-1021). Is it possible for the authors to give some comments on this?

A4. Thanks for the introduction of these papers related to our research. The carbazole-benzonitrile-anthracene, reported in the paper (J. Mater. Chem. C, 2019, 7, 1014-1021), showed EQE over 10%, indicating the radiative-exciton production efficiency over 50% higher than the theoretical limits of the conventional TTU. By DFT calculations, we suppose that this would be due to the fact that the second triplet state (T_2) is higher than the 3 (TT), similar to rubrene. Under this condition, 3 (TT) can be expected to return to two T_1 , because the transition from 3 (TT) to one T_1 requires an instantaneous release of the T_1 energy as vibronic energy. As a result, the radiative-exciton production efficiency can be raised to 62.5% based on the recycling of the triplets. We referred this paper in the revised manuscript.

Q5. Some grammar errors should be modified in the manuscript. For instance, on page 9, line 9, "and thus the contributions".

Q6. In reference No.17, the publication year is wrong. It should be Adv. Funct. Mater. 24, 2064-2071 (2014).

A5 & A6. We are sorry for the careless mistakes. We carefully checked the manuscript again.

Reviewer #2 (Remarks to the Author):

In this manuscript, the authors developed a series of new anthracene derivatives to increase the triplet-triplet upconversion efficiency in OLEDs. With the help of quantum chemical calculations, the authors proposed a possible upconversion mechanism which involves spin-orbit interaction between S_n and T_m . OLED efficiencies of 4.5% and 3.3% were achieved by DMAC- σ -ANTCN and ANT-TRZ, respectively. But notable efficiency roll-off at higher current density was also observed. In this manuscript, the authors tried to prove that the conversion from 3 (TT) to S_n improves the TTU efficiency, but the conclusion is not very convincing since there are several problems contained in this manuscript.

Q1. The ISC quantum yields of the anthracene derivatives in the films are not convincing enough for further calculation of α and η_r . The authors stated in the supporting information (page 34) that k_r and k_{ISC} were assumed to be the same for the solution sample and thin film, but I do not agree with the statement. The energy levels of excited states, especially CT states, are quite sensitive to medium properties. From the photophysical data in Table S3, we can learn that the PLQYs in films were much lower than the PLQYs in solutions, suggesting very different photophysical properties in different medium. The authors need to conduct PL lifetime measurements as well as other characterization in the films to support the calculation.

A1. Thank you for the very valuable comment. We measured the solvent dependence of absorption and PL spectra for the anthracene derivatives as shown in Fig. S2. As reviewer pointed out, these CT states are quite sensitive to medium properties. The ISC rate constants were estimated in benzene solutions, suggesting that their LE characters would be enhanced. The contribution of CT characters in the films would be different from those in benzene solutions. Thus, reviewer's comment is appropriate. We deleted the related sentences and experimental data in the revised manuscript. However, we see no need to change our conclusions because the radiative-exciton production efficiency (η_r) can be evaluated without ISC efficiencies (see Eqs. 4 & S15).

Q2. The mechanism of the spin conversion process in this manuscript is very similar to the "hot" CT channels, but the authors need to clarify how the IC process was blocked in this system. For example, the author stated that the spin conversion process in ANT-TRZ was from T_8 to S_1 (page 13). But according to the calculated energy levels presented in Fig. S3, the energy gaps between T_8 and other triplet excited states (T_2 to T_7) are relatively small. Therefore, the internal conversion should be much faster than the spin conversion process. An explanation is needed.

A2. Thank you for this valuable comment. Our model (see the Supplementary Note 6) is derived from the second-order perturbations including electron-electron and spin-orbit interactions. In our model, the true (perturbed) wavefunction of triplet intermediate ($|^3T_1T_1\rangle$) includes the wavefunctions of $|T_mS_0\rangle$ and $|S_nS_0\rangle$ as the perturbation terms, which represent a pair of molecules in T_n and S_0 and a pair of molecules in S_n and S_0 , respectively. This spin conversion process from $|^3T_1T_1\rangle$ to $|S_nS_0\rangle$ takes place directly without going through the transition to the $|T_mS_0\rangle$. Therefore, there is no internal conversion in our model.

Q3. In addition, the TTU model in the manuscript seems like a combination of TTA and HLCT process, both of which are efficient strategies in OLEDs to exceed the theoretical limit of conventional fluorescence materials. But the device efficiencies in this manuscript are rather low with notable efficiencies roll-off. I think the authors need to reorganize the manuscript and conclude the advantages of the design strategies compared with the reported examples (TTA and HLCT). Perhaps this will shed light on the OLED emitter design. With those changes through a major revision, this work may merit publication in the Nature Communications.

A3. Thank you for the valuable comment. In order to emphasize TTA processes, we fabricated the

OLEDs consisting of DMAC- σ -ANTCN and ANT-TRZ doped into CBP hosts as emitting layers, respectively. As shown in Fig. 5, with an increase of current density, the non-dope devices show efficiency roll-up. On the other hand, the dope devices show no efficiency roll-up, indicating no TTA processes take place due to isolated molecules. These results would support TTA mechanisms in the non-doped OLEDs. Also, regardless of no HLCT for PXZ- σ -ANTCN and ANTTRZ as shown in Fig. S2, we observed higher η_r than 40%. We also agree that HLCT is a key for enhancing photoluminescence efficiency maintaining efficient spin-conversion. However, we suppose that the enhancement of η_r is due to TTA rather than HLCT this time.

REVIEWERS' COMMENTS:

Reviewer #1 (Remarks to the Author):

The authors have addressed all the concerns raised by the referees. The manuscript can be accepted in current form.

Reviewer #2 (Remarks to the Author):

In the revised manuscript, the authors addressed most of concerns of this reviewer, although some of the points may need more experimental evidence from future work. I recommend publication of the revised manuscript, but there are still two minor points can be made to improve the quality of the manuscript.

1. The answer to Q2 is Ok, but it needs some references to support the claim.
2. It would be better to discuss in the text a little more on the advantages of the design strategies compared with the reported examples (TTA and HLCT).

Reviewer #1 (Remarks to the Author):

The authors have addressed all the concerns raised by the referees. The manuscript can be accepted in current form.

We thank the reviewer for the positive comment on our revised manuscript.

Reviewer #2 (Remarks to the Author):

In the revised manuscript, the authors addressed most of concerns of this reviewer, although some of the points may need more experimental evidence from future work. I recommend publication of the revised manuscript, but there are still two minor points can be made to improve the quality of the manuscript.

We thank the reviewer for the positive comment on our revised manuscript.

Q1. The answer to Q2 is Ok, but it needs some references to support the claim.

Thank you for this valuable comment. In the Supplementary Note 6, we added this discussion with some references to support the claim.

Q2. It would be better to discuss in the text a little more on the advantages of the design strategies compared with the reported examples (TTA and HLCT).

Thank you for the very valuable comment. In the revised manuscript, we described the advantages of the proposed mechanism compared with the TTA with the energy level of $^3(TT)$, which is lower than that of a second triplet state (T_2), and the “hot” CT channels through HLCT.

In the revised manuscript, all changes have been highlighted in yellow.